# Effect of Spatial Proximity and Human Thermal Plume on the Design of a DIY Human-Centered Thermohygrometric Monitoring System

**Francesco Salamone** [1,2,*] , **Ludovico Danza** [1] , **Sergio Sibilio** [1,2] **and Massimiliano Masullo** [2]

1   Construction Technologies Institute, National Research Council of Italy (ITC-CNR), Via Lombardia, 49, 20098 San Giuliano Milanese, MI, Italy; danza@itc.cnr.it (L.D.); sergio.sibilio@unicampania.it (S.S.)
2   Department of Architecture and Industrial Design, University of Campania "Luigi Vanvitelli", Via San Lorenzo, 81031 Aversa, CE, Italy; massimiliano.masullo@unicampania.it
*   Correspondence: salamone@itc.cnr.it or francesco.salamone@unicampania.it

**Abstract:** Wearable devices have been introduced for research purposes and especially for environmental monitoring, with the aim of collecting large amounts of data. In a previous study, we addressed the measurement reliability of low-cost thermohygrometers. In this study, we aim to find out how human thermal plume could affect the measurement performance of thermohygrometers. For this purpose, we used a Do-It-Yourself device that can be easily replicated. It consists of 10 iButtons with 3D-printed brackets to position them at different distances from the body. The device was attached to the user's belt in a seated position. We considered two scenarios: a summer scenario with an air temperature of 28 °C and a clothing thermal resistance of 0.5 clo and an autumn scenario with an air temperature of 21 °C and a clothing thermal resistance of 1.0 clo. The results show that the proximity of the measurement station to the body significantly affects the accuracy of the measurements and should be considered when developing new wearable devices to assess thermal comfort. Therefore, we recommend that at least two thermohygrometers be considered in the development of a new wearable device if it is to be worn on a belt, with one positioned as close to the body as possible and the other at least 8 cm away, to determine if and how the standard thermal comfort assessment differs from the user's personal perception and whether spatial proximity might also play a role.

**Keywords:** thermal comfort; IoT; DIY; wearables; spatial proximity; human thermal plume; monitoring system

## 1. Introduction

Technological progress over the last two decades has helped to change how many problems are handled in all areas of knowledge and has made available new tools and devices [1]. One example of this technological progress is the so-called Internet of Things (IoT), which is defined as a network of interconnected devices that are easy to communicate with and often inexpensive or portable. The IoT approach has enabled the web to evolve from the static websites of the 1990s to the Web 2.0 (Social Networking Web) of the 2000s to the Web 3.0 (Ubiquitous Computing Web) of the present [2,3]. This last iteration enables the user to become the maker of open, low-cost hardware technologies that can be used in any technological domain and has enabled the spread of the Do-It-Yourself (DIY) approach [4,5]. Anyone can apply the principles of the Do-It-Yourself and IoT philosophy with technologies such as Arduino, a series of open-source hardware-based microcontrollers equipped with digital and analogue input/output pins that can be connected to various sensors or expansion boards, or Raspberry Pi, a series of small single-board computers designed primarily to learn programming in order to create smart, low-cost solutions on a DIY basis and share them with other makers and researchers everywhere.



The DIY paradigm is on the rise, and the reasons for using this movement have been explored in [6]. A holistic view of DIY with the IoT was also explored in [2]. With the increasing number of smart devices, we are also gradually entering the era of smart IoT [7,8], where large amounts of data collected from "smart things" are used to train Machine Learning (ML) algorithms [9] or more generally to develop artificial intelligence (AI) applications. The use of IoT devices is also increasing in building control. As highlighted in [10], monitoring with high spatial and temporal resolution has been identified and can be integrated with human-centered and automated control measures to improve thermal comfort and automated control methods. In this sense, IoT sensing provides more environmental information with greater spatial granularity compared with traditional sensing [11] and could have far-reaching implications in determining the relationship between human health and environmental quality [12] due to its ubiquitous intelligence [13].

In this context, it becomes clear that there is a need to consider affordable and accurate, low-cost solutions for monitoring environmental variables that can be used as an alternative to expensive hardware to understand how environmental variables (e.g., air temperature and relative humidity) can affect the comfort levels of different occupants during their daily lives.

### 1.1. Thermohygrometers in the Built Environment: Field of Application

Thermohygrometers, which combine a thermometer to measure air temperature and a hygrometer to measure humidity in a single device, are currently used for research purposes and often take advantage of low-cost sensor technology. For example, in a recent study, they were used in physical environments to improve the user experience [14]; in another study described in [15,16], thermohygrometers were used in combination with other sensors in a framework to assess user-perceived thermal comfort using ML. Thermohygrometers have also been used in combination with other sensors to assess indoor thermal comfort for people with health problems [17]. In [18], an iButton DS1923 [19] was placed slightly above the ankle with the sensor side facing outward and used with other sensors capable of collecting physiological data to train a supervised ML model. In [20], the iButton DS1925 was used to monitor air temperature near the body: in this sense, authors used the acronym "$t_{nb,w}$" to identify the "wrist near body temperature". To monitor $t_{nb,w}$, they used a 3D-printed clip that was attached to the Fitbit Versa band. A similar approach was taken in [21].

In outdoor applications, they have been used to collect data from a pedestrian perspective [22,23] to monitor microclimatic conditions in cities, e.g., to investigate how different urban configurations and architectural designs may affect urban microclimates or to understand the impact of the Urban Heat Island (UHI) in the hottest seasons and the human wind chill during the coldest period. However, assessing human responses to indoor and outdoor environments requires considering many interacting factors [24] and behavioral and cognitive aspects [25–27]. Moreover, it should be considered that the comfort perception of building occupants is influenced by the simultaneous presence of several environmental stimuli, i.e., visual, thermal, acoustic, and air quality [28]. Studies on combined factors in the laboratory are limited because the experimental setup is complex and time-consuming, and the hardware to monitor the different required environmental variables simultaneously is expensive [29]. Low-cost solutions could represent an alternative to help study new user comfort and health frameworks in real and multisensory contexts, even though uncontrolled, using a human-centered approach. This approach is recently gaining popularity in the scientific community as it allows the consideration of multiple factors related to human sensations and complex mental states that interact in the perception of thermal comfort [30].

*1.2. Thermal Exchange between Humans and the Environment*

A human body exchanges heat with the environment through various effects:

- radiation, the primary way the body exchanges heat with the environment; when the air temperature is lower than the skin temperature, the body radiates heat in the form of electromagnetic waves;
- convection, due to convective motion between the body and the surrounding air (or water);
- conduction, typically when a body is in contact with another object, such as a chair;
- respiration, in which a person exhales air that is usually warmer than the surrounding air and releases a small amount of heat.

The human thermoregulatory system takes care of body temperature and, in cold environments, activates mechanisms such as shivering and constriction of blood vessels to generate heat, while in hot environments, it activates sweating, which is the evaporation of fluids through the skin, or it activates the dilation of blood vessels to release heat.

The heat balance between environmental variables and the human body is shown in [31] where the skin temperature and sweat rate are considered the most important physiological data influencing the heat balance [31].

The simple human thermal model considers the skin surface as the boundary between the thermal environment and the human metabolic heat which moves for conduction and blood flow from the interior to the skin surface [32]. Other thermal models are provided in [31,33].

The constant heat exchange between the human body and the environment is commonly known as Human Thermal Plume (HTP) which, in an indoor environment with a calm atmosphere, generates a laminar flow on the lower part of the body which then becomes turbulent at the chest level, with the thickness of the plume variable from 2 mm at the knee level to 20 cm at the chest level and an air velocity that could reach 0.3 m/s at 50 cm from the head level [34].

The influencing factors of HTP are:

- temperature gradient between the human body and air temperature, with an increase in the intensity of HTP in terms of thickness and air velocity with an increase in the thermal gradient;
- body posture, with a wider thickness of HTP for an indoor occupant with sedentary activity and lower ascending mean velocity;
- type of clothing, with a sensible reduction of the intensity of HTP when wearing loose clothing due to the thermal insulation effect.

*1.3. Motivation of the Proposed Study Based on the Previous Consideration*

To improve the possibility of using low-cost devices in living lab studies focusing on humans, a previous study [35] has defined a procedure to evaluate the performance of seven different low-cost thermohygrometers in a controlled environment in order to understand the differences in determining classical indices of indoor/outdoor thermal comfort and in assessing thermal stress, showing good behavior. Following the conclusions of the previously mentioned work [35], in this article, we want to analyze how Spatial Proximity (SP), i.e., the impact or influence area of an object [36,37], or the human body in this specific case, could affect the monitoring of air temperature and relative humidity under the effect of the HTP. To this end, we developed a DIY solution based on 3D-printed parts and a carbon rod used to position 10 different iButtons DS1923 [19], an instrument for controlling temperature and relative humidity, whose performance and deviations from each other were also characterized in a test in a climatic chamber. A user wore the system for two days in two different configurations: a typical summer day indoors and a typical autumn day indoors. The clothing corresponding the two seasonal conditions was: 1 clo for the autumn configuration and 0.5 clo for the summer configuration; in both

scenarios, a metabolic rate of 1 met was considered coherent with a relaxed office activity (seated quietly) [38].

The environmental data are used together with subjective parameters to define the classical index of thermal comfort and the Predicted Mean Vote (PMV), which identifies the mean value of votes of a group of occupants on a seven-point thermal sensation scale [39].

The following sections describe the materials and methods, the results of the test, a discussion section that also identifies the limitations of the proposed study, and the conclusion section that outlines the main findings of this study and some future improvements that could be considered.

## 2. Materials and Methods

### 2.1. Calibration of Ten iButtons DS1923

Ten iButtons DS1923 sensors were considered as a reference thermohygrometers. Each iButtons DS1923 has a size of 5.89 mm (H) and 17.35 mm (D) and a measurement range of −20 °C to +85 °C for air temperature (T) and from 0 to 100% for Relative Humidity (RH). The iButton DS1923 has a tiny opening in its lid covered by a special filter that allows water vapor to pass through and reach the internal humidity sensor. The resolution is selectable: 8-bit, which corresponds to a resolution of 0.5 °C for T and 0.64% for RH, or 11-bit, which corresponds to 0.0625 °C for T and 0.04% for RH. The operative temperature can be between −20 °C and +85 °C. To evaluate the behavior of the sensors and to correct difference in monitoring T and RH, the ten iButtons DS1923 were placed in a GENVIRO-030 climatic chamber (Figure 1), which allows control of temperature and relative humidity in the internal volume of 30 L in the ranges −40 to 180 °C and 10 to 98%, respectively. The temperature profile used in the test, which lasted approximately 2 days, was between −10 and 40 °C. The relative humidity was between 20 and 90%. The T and RH ranges were consistent with most previously conducted studies [40]. The volume containing the ten iButtons was small enough (LxWxH = 75 mm × 55 mm × 5.89 mm) to avoid inhomogeneous distribution of air temperature and relative humidity in the chamber, as practically demonstrated in previous experiences [35].

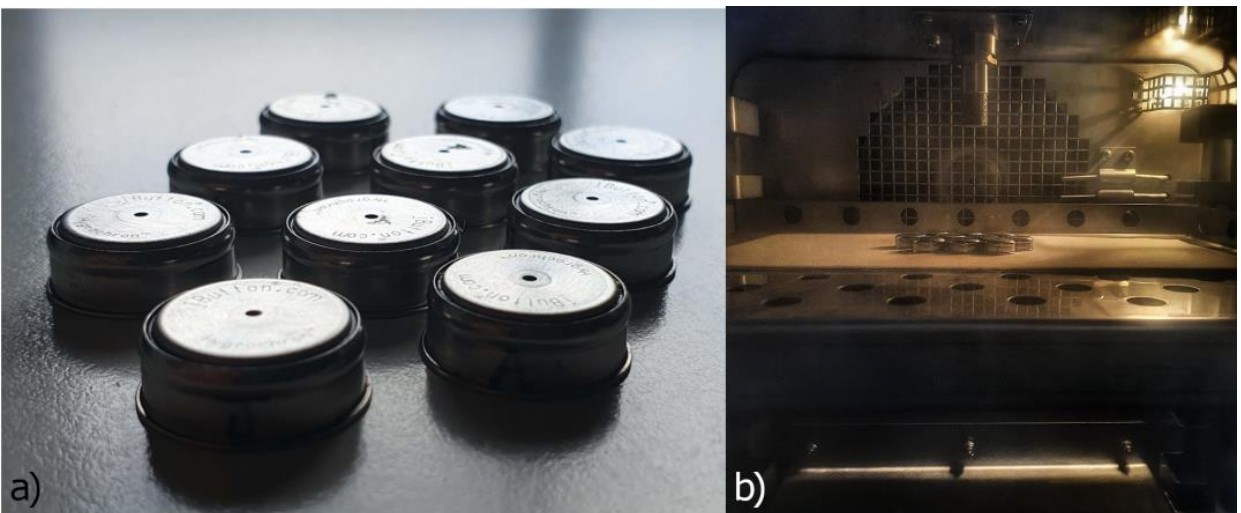

**Figure 1.** iButtons DS1923: (**a**) detail of the ten iButtons DS1923; (**b**) the ten iButtons DS1923 as installed in the climatic chamber.

### 2.2. DIY-Based Approach to Construct the Hardware

The hardware elements of the system were created with a 3D printer according to the DIY philosophy. A clip and 10 housings for the ten iButtons DS1923 sensors were designed and 3D printed (Figure 2a). The 3D-printed parts and the ten DS1923 sensors whose performance was checked in the climatic chamber were arranged along a carbon rod

of 5.5 mm in diameter and 64 cm long (Figure 2b) and placed equidistant 8 cm from each other, except for the first segment on the left in Figure 2b, where a double resolution was considered, with an additional sensor placed 4 cm from the left end to account for different boundary situations (greater proximity to the human body).

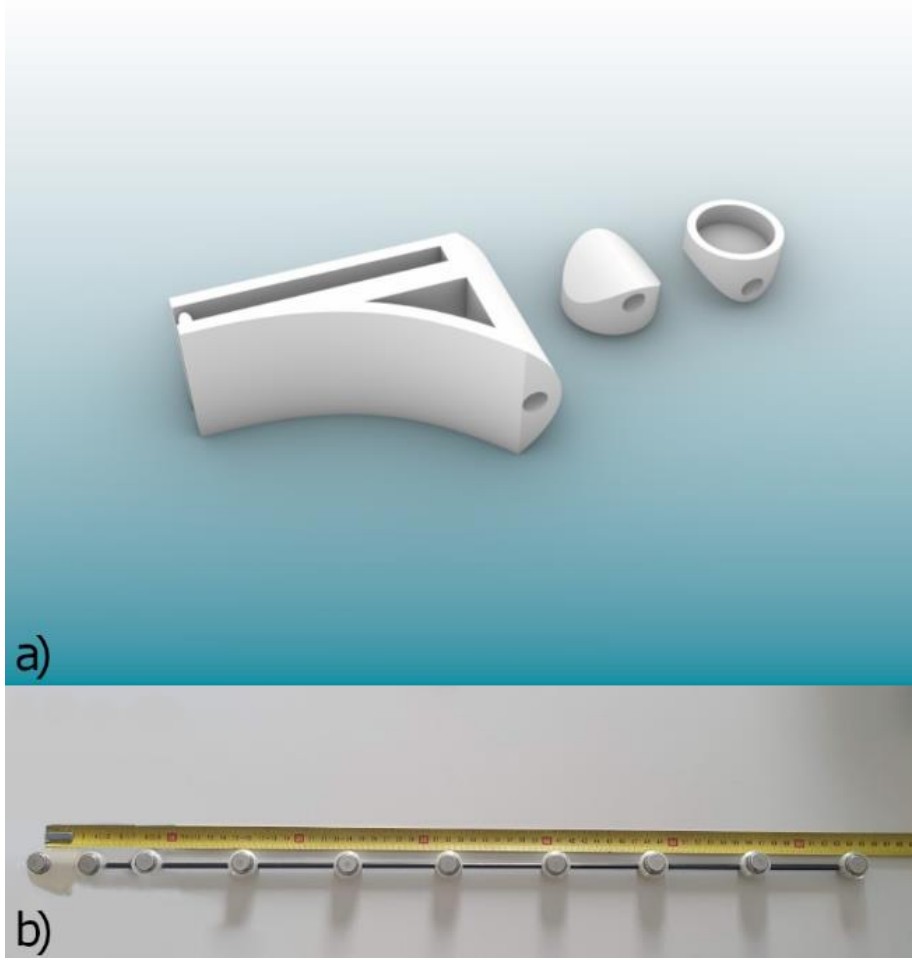

**Figure 2.** Hardware used for the test: (**a**) design of 3D-printed parts; (**b**) the ten iButtons DS1923 as mounted on 3D-printed parts with carbon rod.

The first point on the left side of Figure 2b was used to measure the air temperature closest to the human body. Following the nomenclature used in [20], we defined it as the "waist near body temperature" and labeled it "$t_{nb,w}\_A$". The other points are named with a label such as "x cm_y", where x represents the distance in centimeters from the left end, or the human body, while y is the letter assigned to the iButtons, according to the test performed in Section 2.1.

### 2.3. Testing Scenarios Considered

The system shown in Figure 2b was placed on a user's belt, in line with the "abdomen level" indicated in the ISO 7726 [35], corresponding to a height of approximately 0.6 m when sitting and 1.1 m when standing, on two different days: 8 and 14 November 2022, considering two scenarios:

- Autumn, with an air temperature in the range of 21 °C, a thermal resistance of clothing of 1.0 clo, a real office scenario with a user who could move freely out of the office (8 November 2022)

- Summer, with an air temperature in the range of 28 °C, a thermal resistance of clothing of 0.5 clo, an office scenario reproduced in the ZEBlab [41] with a user enclosed in an indoor space (14 November 2022)

Figure 3 shows the two experimental setups with the environments considered, the user position, and the hardware worn with a detail of the developed system as fitted by the user.

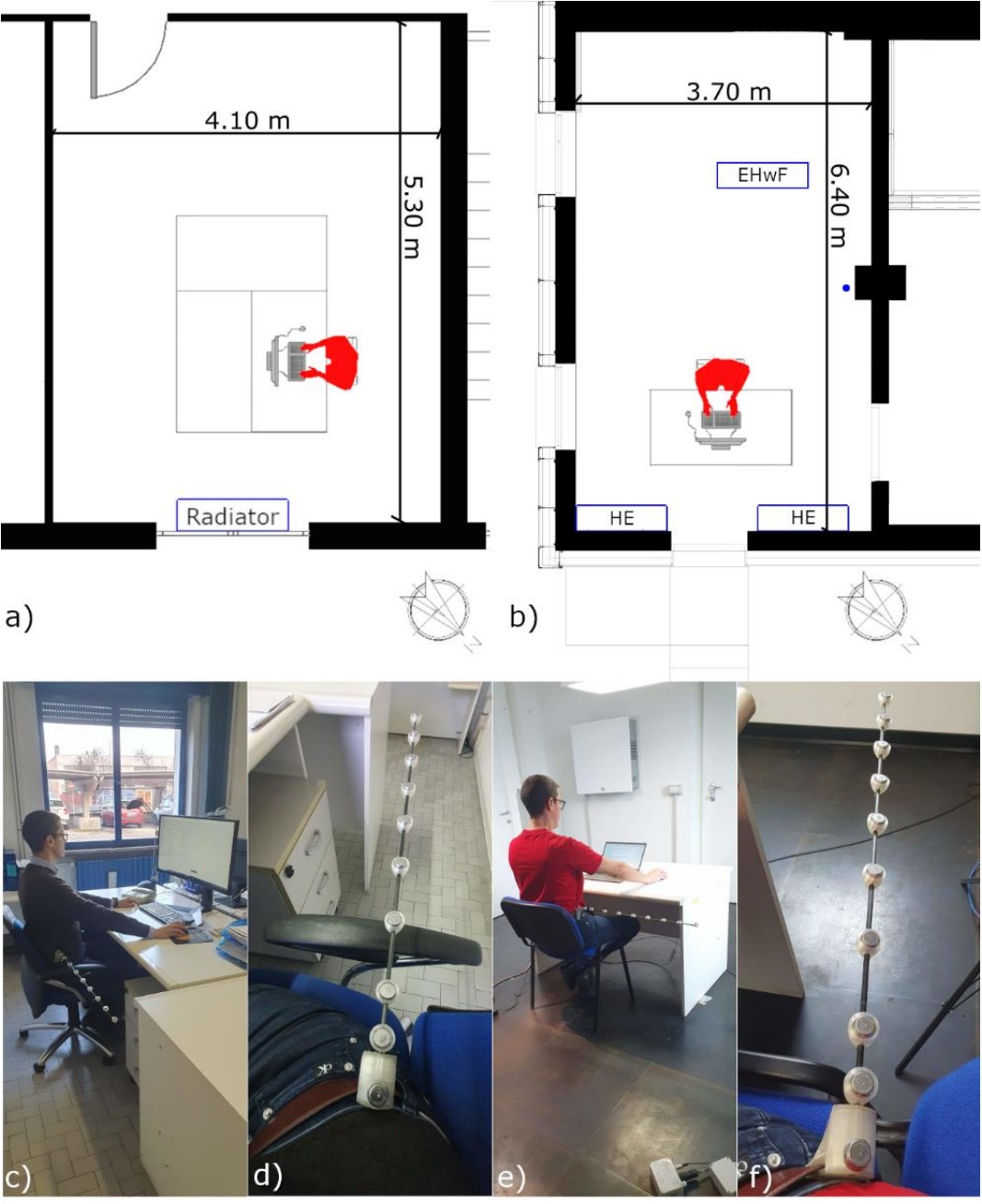

**Figure 3.** Autumn and summer setup: (**a**) office plant used for the autumn scenario; (**b**) ZEBlab room used for simulating the summer scenario (HE = Heat Exchanger; EHwF = Electric Heater with Fan; with blue dot, the position of Pt100 connected to the thermoregulator used to control the EHwF); (**c**) picture of the autumn scenario with user and hardware in the considered environment; (**d**) detail of the hardware as mounted in the autumn scenario; (**e**) picture of the summer scenario with user and hardware in the considered environment; (**f**) detail of the hardware as mounted in the summer scenario.

In the autumn scenario, heating was provided by one radiator placed under the window (Figure 3a). In the summer scenario, the highest temperature of approximately 28 °C was achieved by using an electric heater with fan (Figure 3b). It was controlled by a thermoregulator, the Vemer HT NiPT-1, which guarantees a temperature setpoint of 28 °C (±0.5). The thermoregulator was connected to a platinum resistance Pt100 positioned near the right wall of the room (blue dot in Figure 3b) that could provide the air temperature. In the autumn scenario, ventilation was provided by the manually opening of doors and windows. In the simulated summer scenario in the ZEBlab, ventilation was provided by mechanical ventilation with two heat recovery units that allowed a complete air exchange per hour.

In both scenarios, the rooms were occupied by the tester alone wearing an Empatica E4 wristband on the left arm. In the autumn scenario, the wearable system composed of the ten iButtons DS1923 sensors was worn on the opposite side of the window glass, although the environment was not characterized by direct sunlight during the day and test hours. In the simulated summer scenario, on the other hand, the wearable system was worn on the window side, as this was characterized by triple glazing with low emissivity and, during the test the position where the user and the ten measurement points were located, was not characterized by direct sunlight. The additional light sources are fluorescent tubes in the case of the autumn scenario and LED in the case of the summer scenario.

Even though it was possible to perform Human Activity Recognition (HAR) using semi-supervised techniques [42] (i.e., sitting, walking, etc.), the participant used the tag button available on the E4 to indicate both the beginning of a walking activity and the end of the single walking activity. In this way, we could identify the period of movement (with a red line in Figure 3) and the period the user was not sitting in the office (gray-filled area in Figure 3) considered for the autumn scenario. The Empatica E4 also provided us with the ability to record skin temperature, which was then correlated with the air temperature measured by the 10 iButtons to check for a possible correlation. We used the wrist to monitor skin temperature because, as shown in [43], it is linearly correlated with the mean skin temperature calculated using Hardy & Dubois's seven-point method [44] and because the extremities of the human body (sole, foot, arm, and hand) are more sensitive to the thermal environment [45]. However, as demonstrated in [46], the skin temperature distribution across the whole body varies slightly when considering a neutral setting (defined by the subjects) or in warm (slightly above 30 °C) testing conditions.

In both scenarios, all data from the ten iButtons DS1923 sensors were collected at a sampling rate of 1 min and merged into a single database. A Python script was used for the analysis described in Section 3, with Seaborn [47] and Matplotlib [48] for visualization, SciPy [49], typically used in combination with Numpy [50], a package used in this case to calculate the coefficient of determination $R^2$ [51], and the Scikit-learn metric to calculate the RMSE [52]. Pandas [53] was used to enable the data structure deployment, while Pythermalcomfort [54,55] was used to calculate the PMV [56,57], the most commonly used index for determining thermal comfort (TC) in indoor spaces equipped with HVAC systems or naturally ventilated and which depends on air temperature and relative humidity values, among other subjective and objective variables [15].

## 3. Results

### 3.1. Comparison in a Controlled Environment of the T Values Monitored by the 10 Sensors

The following are the results of the first test carried out in the climatic chamber to evaluate the behavior of the sensors and the correct difference in relation to T monitoring. Figure 4 shows the lineplot of the T values of the i-th sensor compared with the average trend. The histplot as well as the boxplot of the absolute difference between the value measured by each of the ten sensors and the average reference value are located on the right-hand side.

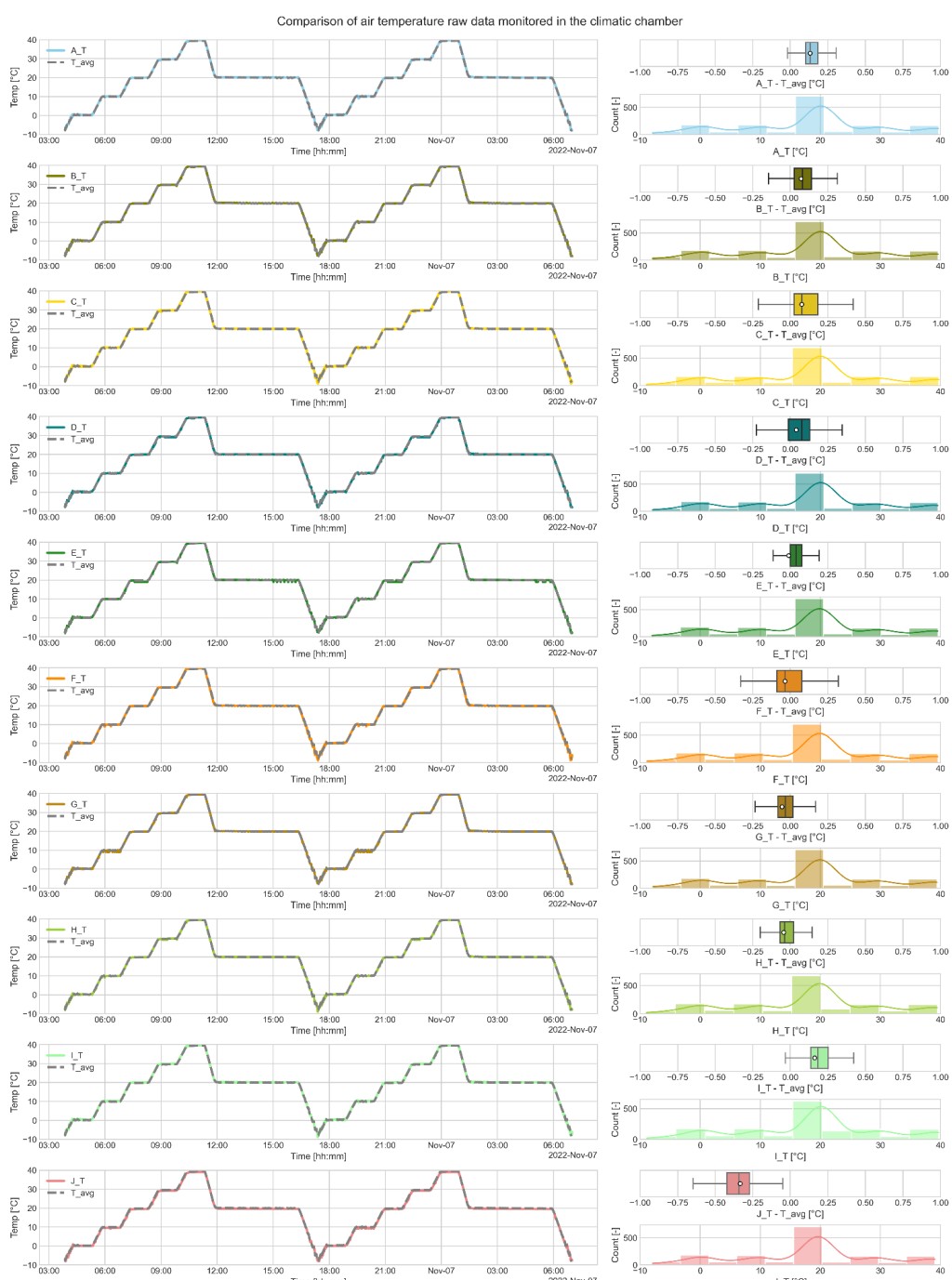

**Figure 4.** T raw data—comparison in the climatic chamber: on the left, the lineplot of the i-th sensor when compared with the average values; on the right, the histplot and the boxplot of the absolute difference between values recorded by each sensor and the average values (the white dot indicates the mean value and the black line indicates the median).

Looking at the values monitored by the ten sensors, the results show that the monitored T are almost constant, with the deviations on average tall limited to a range between −0.5 °C and 0.5 °C, although the extreme values can diverge by more than 0.5 °C, as in the case of the J_T sensor.

We can correct for the very small difference by applying a simple linear regression over the first 50% of the data. Table 1 shows the linear regression coefficient (m) and the intercept (q) considered for all sensors.

**Table 1.** T values—Linear regression. Slope (m) and intercept (q) for all sensors from A to J.

|   | A | B | C | D | E | F | G | H | I | J |
|---|---|---|---|---|---|---|---|---|---|---|
| **m** | 0.9988 | 1.0001 | 0.9934 | 1.0044 | 1.0004 | 0.9946 | 1.0007 | 1.001 | 0.9978 | 1.0053 |
| **q** | −0.104 | −0.07 | 0.0232 | −0.1031 | 0.0161 | 0.1032 | 0.0544 | 0.0273 | −0.1339 | 0.2521 |

The new comparison with the remaining 50% of data linearized (Figure 5) confirms the possibility of using these 10 sensors in the further course of the experiment without considering a discrepancy in the T measurements due to the possible intrinsic error of the instruments used.

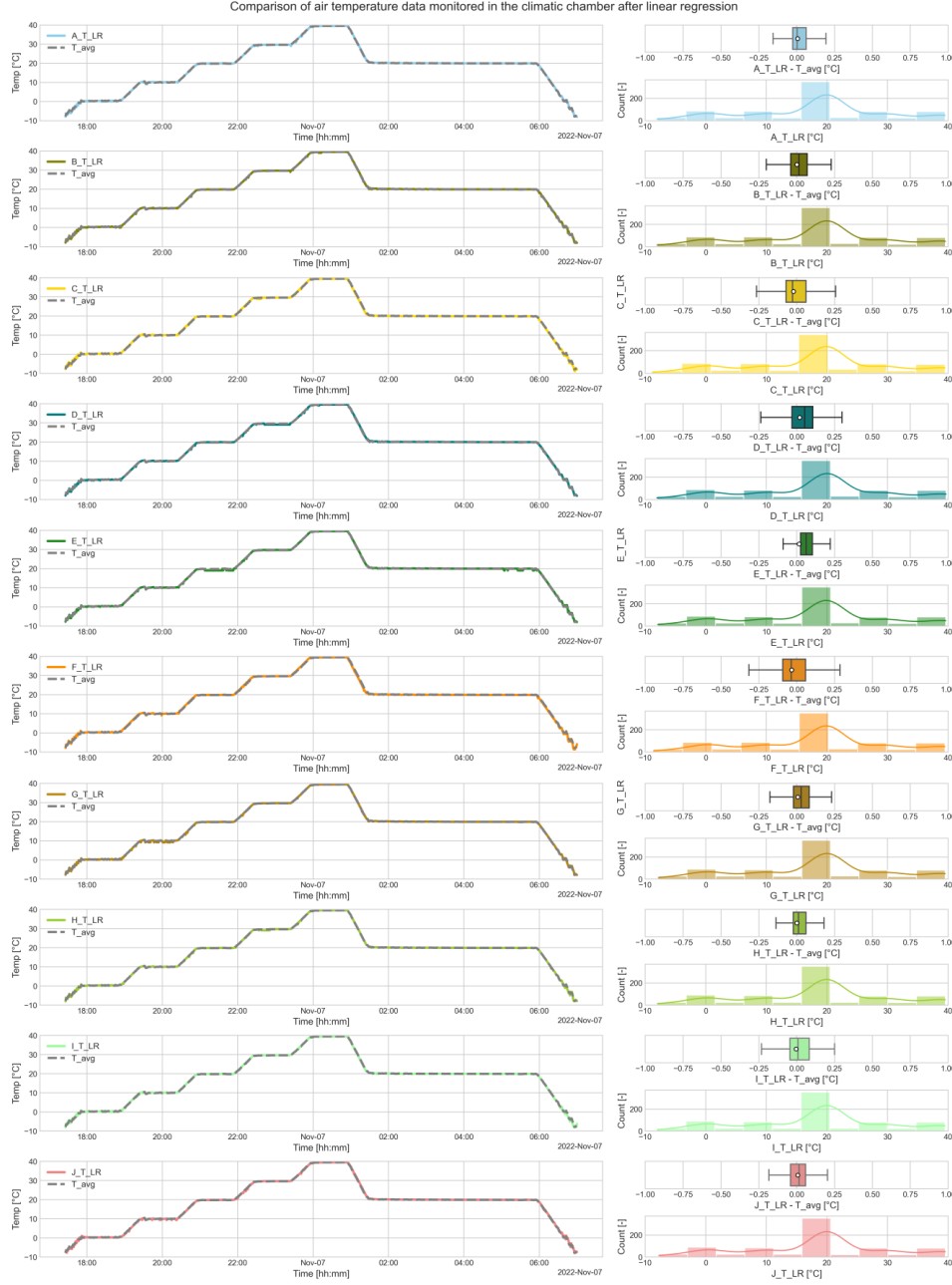

**Figure 5.** T data after linear regression (LR)—comparison in the climatic chamber: on the left, the lineplot of the i-th sensor when compared with the average values; on the right, the histplot and the boxplot of the absolute difference between values recorded by each sensor and the average values (the white dot indicates the mean value and the black line indicates the median).

### 3.2. Comparison in a Controlled Environment of the RH Values Monitored by the 10 Sensors

The same identical approach was used to check the behavior of the sensors and correct the difference in terms of RH monitoring. Figure 6 shows the lineplot of the RH values of the i-th sensor compared with the average trend.

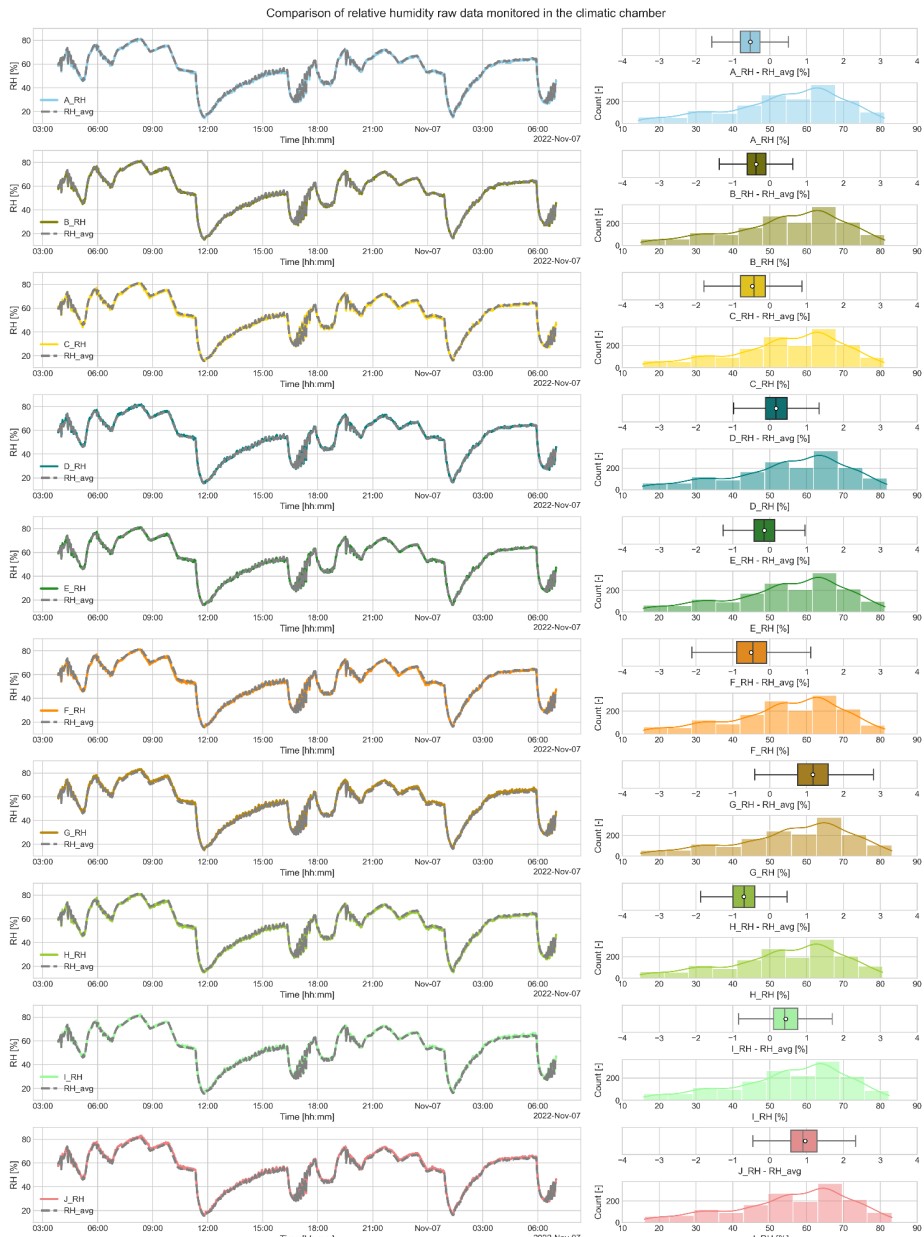

**Figure 6.** RH raw data—comparison in the climatic chamber: on the left, the lineplot of the i-th sensor when compared with the average values; on the right, the histplot and the boxplot of the absolute difference between values recorded by each sensor and the average values (the white dot indicates the mean value and the black line indicates the median).

Even though the discrepancy between each sensor and the average values is very small and corresponds to the standard ISO 7726 [35], we would like to reduce the difference between the sensors as much as possible. For this reason, we also performed a linear regression in this case over the first 50% of the data. Table 2 shows the linear regression coefficient (m) and the intercept considered (q) for all sensors.

**Table 2.** RH values—Linear regression. Slope (m) and intercept (q) for all sensors from A to J.

|       | A      | B      | C       | D      | E       | F       | G      | H      | I       | J       |
|-------|--------|--------|---------|--------|---------|---------|--------|--------|---------|---------|
| **m** | 0.9954 | 1.0007 | 1.0127  | 0.9912 | 1.0045  | 1.0192  | 0.9753 | 0.9968 | 1.01    | 0.9832  |
| **q** | 0.766  | 0.3203 | −0.1757 | 0.2963 | −0.1073 | −0.4822 | 0.2039 | 0.8712 | −0.9748 | −0.0579 |

Figure 7 shows the results in relation to the remaining 50% of RH data linearized and confirms the possibility of using these 10 sensors in the further course of the experiment without considering a discrepancy in the measurements of RH due to the possible intrinsic error of the instruments used.

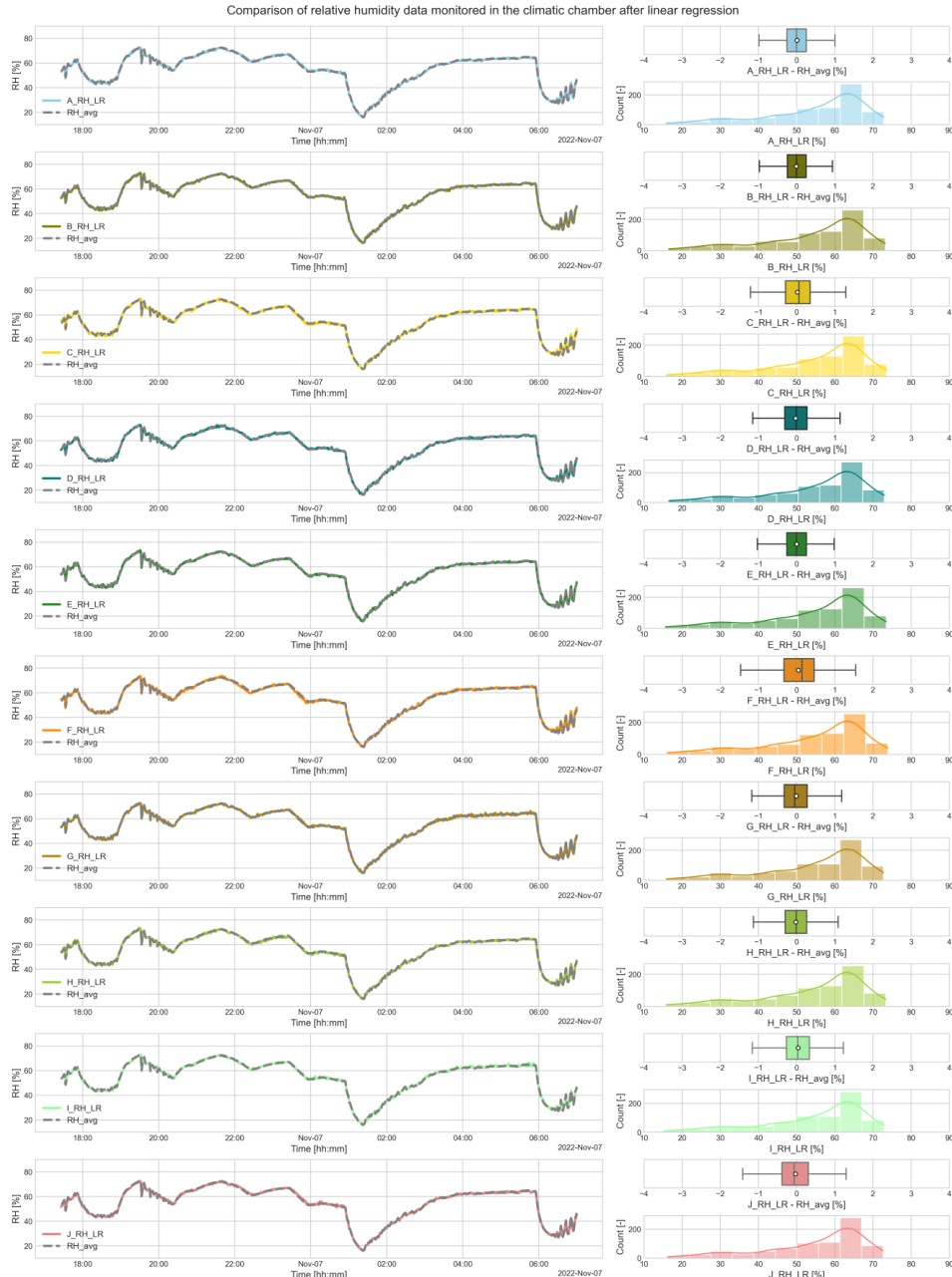

**Figure 7.** RH data after linear regression (LR)—comparison in the climatic chamber: on the left, the lineplot of the i-th sensor when compared with the average values; on the right, the histplot and the boxplot of the absolute difference between values recorded by each sensor and the average values (the white dot indicates the mean value and the black line indicates the median).

### 3.3. Comparison in Real Autumn Scenario of the Air Temperature and Relative Humidity of the 10 Sensors

Figure 8 shows the lineplots of the T data for the real autumn scenario. Since Empatica E4 offers the possibility to tag events, we tagged all movements (shown with a vertical red line in the plot), while the gray-filled areas in Figure 8 are those where the user is not in the office used for the test.

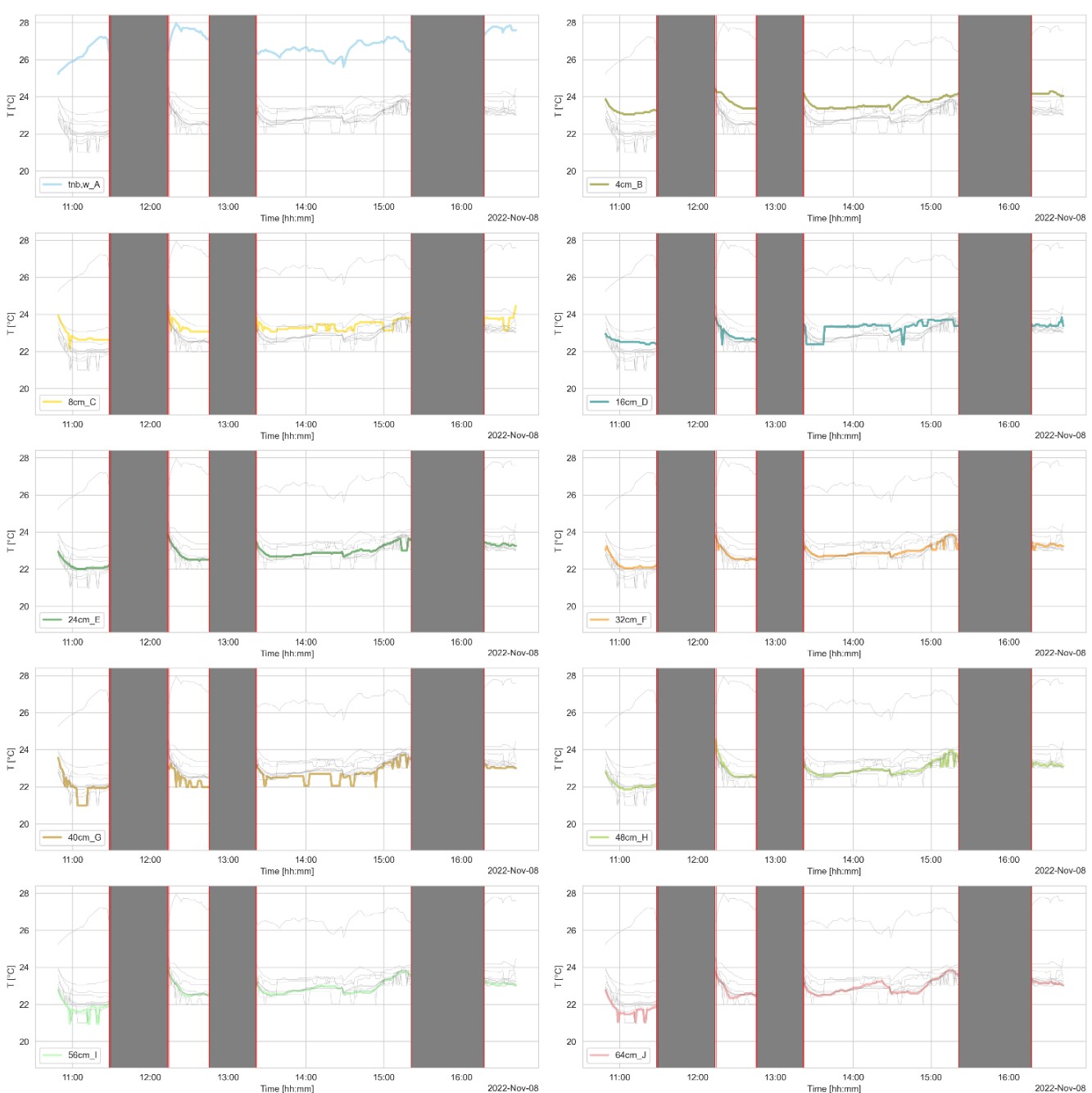

**Figure 8.** T data comparison in infield autumn scenario.

The line graph at $t_{nb,w}$_A is very different from all the other curves. Since the Empatica E4 also records skin temperature at a sampling rate of 4 Hz (4 measurements per second), we averaged all the minutes and used the list of data of skin temperature data for comparison with the data recorded by the ten iButtons. Figure 9 shows the results and is constructed to show:

- on the diagonal, the distribution of each variable;
- at the bottom of the diagonal, the bivariate scatter plots with a fitted line defining the correlation between the two variables (when there is a strong positive correlation between the variables, the points tend to form a line increasing from left to right, i.e., when the value of one variable increases, the value of the other variable also increases, and vice versa for negative correlation);
- at the top of the diagonal, the value of the Pearson correlation coefficient ®and the significance level *p*-value as asterisks ('' if $p > 0.05$, '*' if $p \leq 0.05$, '**' if $p \leq 0.01$, '***' if $p \leq 0.001$).

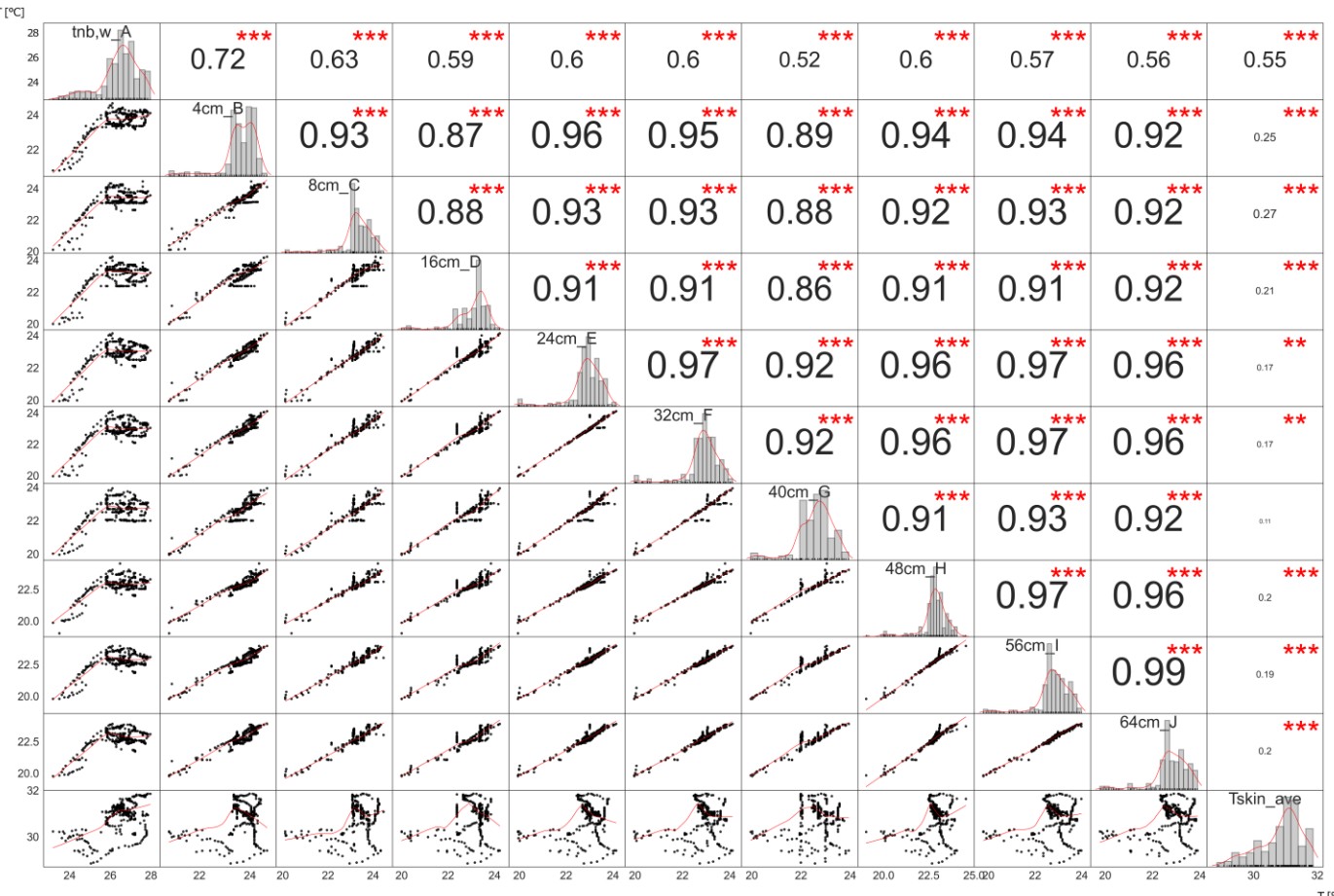

**Figure 9.** T—environmental and skin data pairwise relationships in infield autumn scenario; scatter plots in the lower triangle; the histplot and Kernel Density Estimation line of the i-th sensor in the diagonal; Pearson correlation coefficient in the upper triangle with *p*-value ('' if $p > 0.05$, '*' if $p \leq 0.05$, '**' if $p \leq 0.01$, '***' if $p \leq 0.001$).

At the bottom of the diagonal, except for the first row and the last column, it can be observed that the r coefficient is at least 0.86 or higher, showing a strong [58] correlation between the different positions of the air temperature measurement in line with the physical nature of thermal conduction and convection. In fact, the bivariate scatter plots at the bottom of the diagonal, except for the first column and the last row, show a linear correlation for all measurements while the comparison of the Tskin values with the measurements of all other sensors in the form of bivariate scatter plots (last row) or r and *p* values (last column) shows a moderate positive correlation with the air temperature measured at $t_{nb,w}$_A, which already becomes poor at 4 cm. This shows how, in accordance with the concept of HTP, the Tskin could have an influence on the thin layer of air near the human body, while it cannot influence the other layers at a great distance, as evidenced by the very low r coefficient. The statistical significance of the results is guaranteed by the very low *p*-values.

In Figure 10, we observe the trend of RH in the autumn scenario.

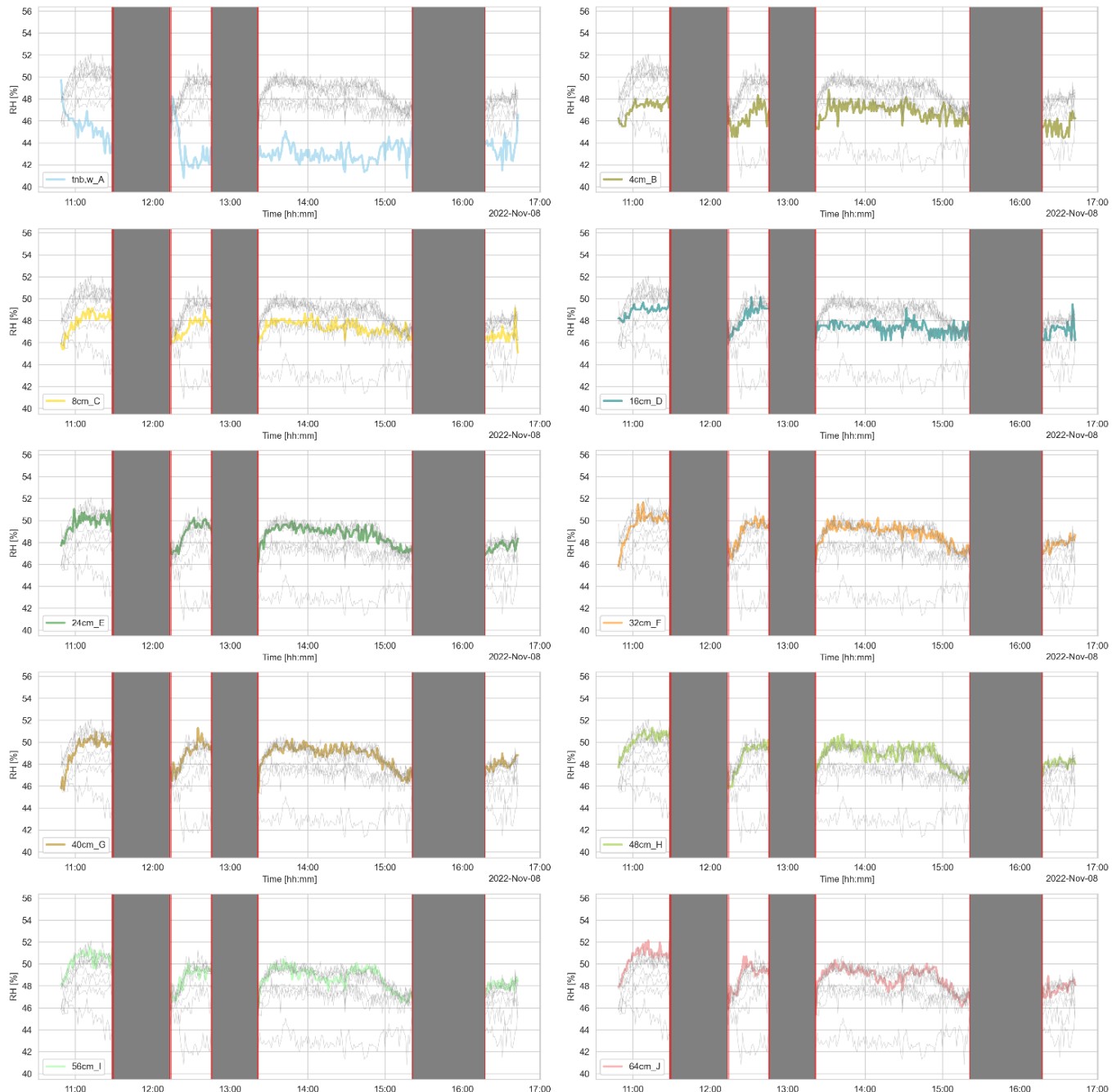

**Figure 10.** RH data—comparison in infield autumn scenario.

At $t_{nb,w}\_A$, the effect of HTP in relative humidity monitoring is more relevant than at any other distance. In this sense, the higher temperature causes a reduction in relative humidity as a concomitant effect, as was found in another, earlier experiment where a thermohygrometer was positioned closest to a heat source, causing a drastic rise in temperature above room temperature and a drastic drop in relative humidity [59]. The same is likely to have occurred in this case with a heat source, the human body, causing a rise in temperature and a consequent drop in relative humidity. The data show that any water vapor released by the human body is not sufficient to counteract the temperature rise due to proximity with a hot body at a higher temperature.

### 3.4. Comparison in Simulated Summer Scenario of the Air Temperature and Relative Humidity of the 10 Sensors

The same situation can be verified in the summer simulated scenario (Figure 11). The only difference in this case is that the lineplot at $t_{nb,w}$_A is on average 4 °C higher than the other air temperatures, whereas the average difference in the previous case is approximately 3 °C.

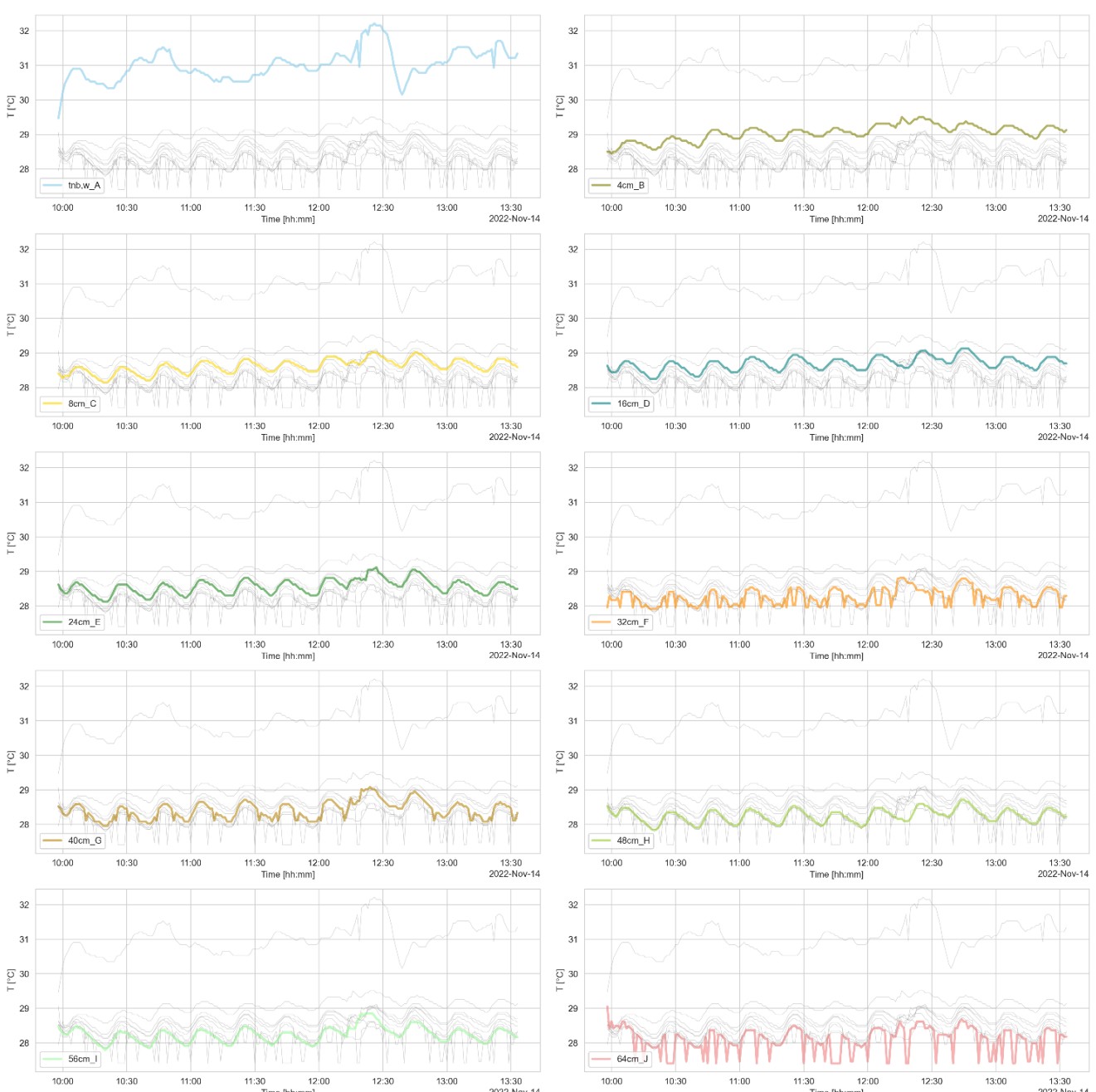

**Figure 11.** T data—comparison in infield summer scenario.

As for the autumn scenario, we used the list of data of skin temperature monitored by Empatica E4, averaged each minute, and then compared this information with the data recorded by the ten iButtons. Figure 12 shows the results in terms of the Pearson correlation coefficient (r) and the *p*-value ('*' if $p \leq 0.05$, '**' if $p \leq 0.01$, '***' if $p \leq 0.001$).

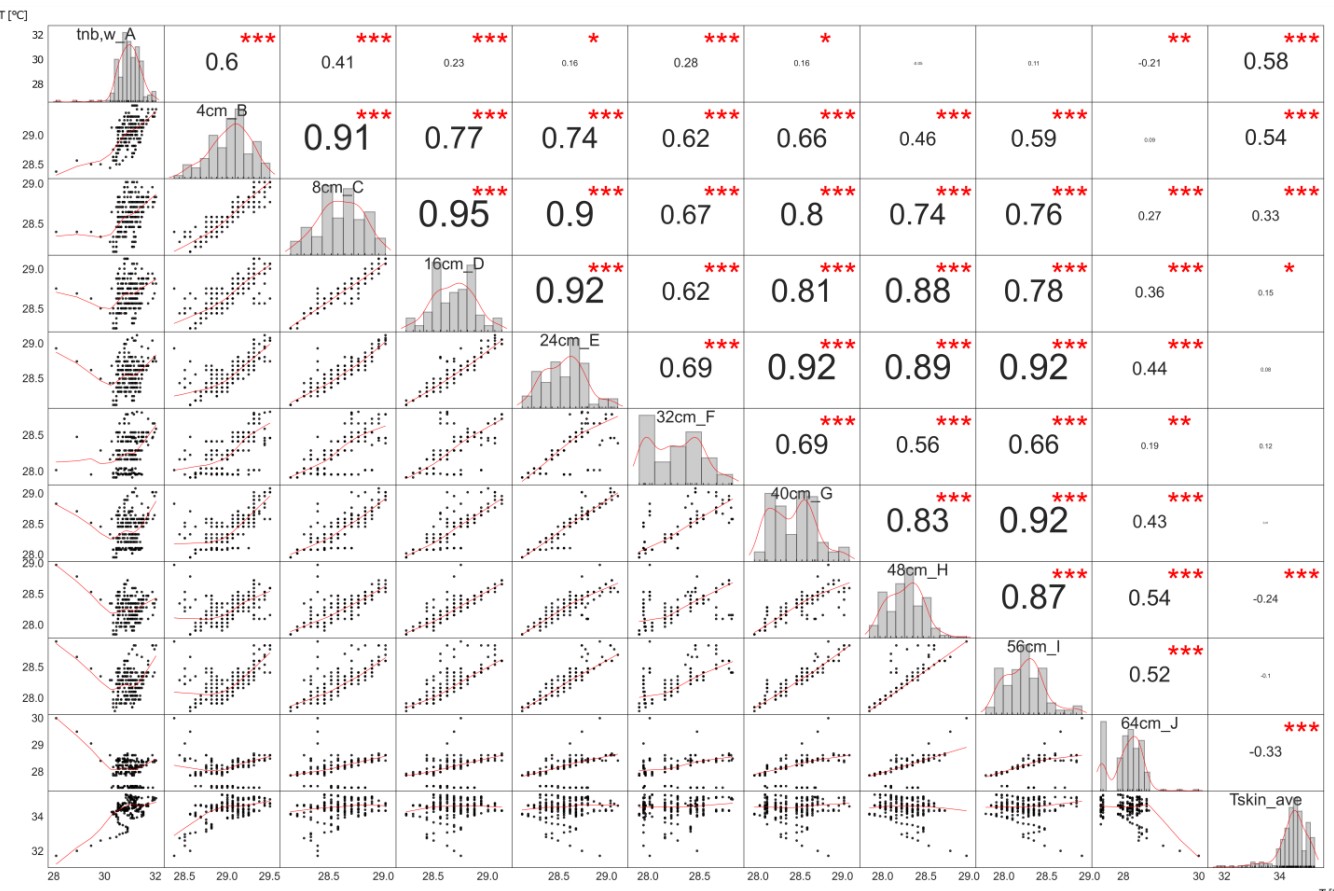

**Figure 12.** T—environmental and skin data pairwise relationships in infield summer scenario; scatter plots in the lower triangle; the lineplot and the histplot of the i-th sensor in the diagonal; Pearson correlation coefficient in the upper triangle with *p*-value ('' if $p > 0.05$, '*' if $p \leq 0.05$, '**' if $p \leq 0.01$, '***' if $p \leq 0.001$).

Unlike the autumn scenario, in this case, on the bottom of the diagonal, except for the first row and last column, it can be observed how the r coefficient is more variable, showing a great influence on the point of measurement of air temperature when using a mixed system composed of an electric heater with fan and two heat exchangers. In fact, as confirmed by the bivariate scatter plots, on the bottom of the diagonal, except for the first column and last row, the overall behavior is extremely wide, especially if compared with the trend of Figure 9, where, instead, the plots are perfectly aligned along the red line. Regarding the comparison of the Tskin values with the measurement of all the other sensors, in terms of r coefficient and *p* values (last column), we can see how there is a statistically significant moderate positive correlation with air temperature measured at $t_{nb,w}$_A and 4 cm which becomes poor as early as 8 cm. In terms of bivariate scatter plots (last row), we can observe how, except for the first two graphs on the left, the trend of the red line is horizontal, demonstrating how air temperature monitored in the different positions is poorly correlated and independent from skin temperature.

In Figure 13, we observe the trend of RH in the summer scenario. On the contrary to what might be expected (because we normally assume that the body sweats more at higher temperatures), the difference between the RH curve at $t_{nb,w}$_A is much less pronounced than in the other curves, even compared with the previous case.

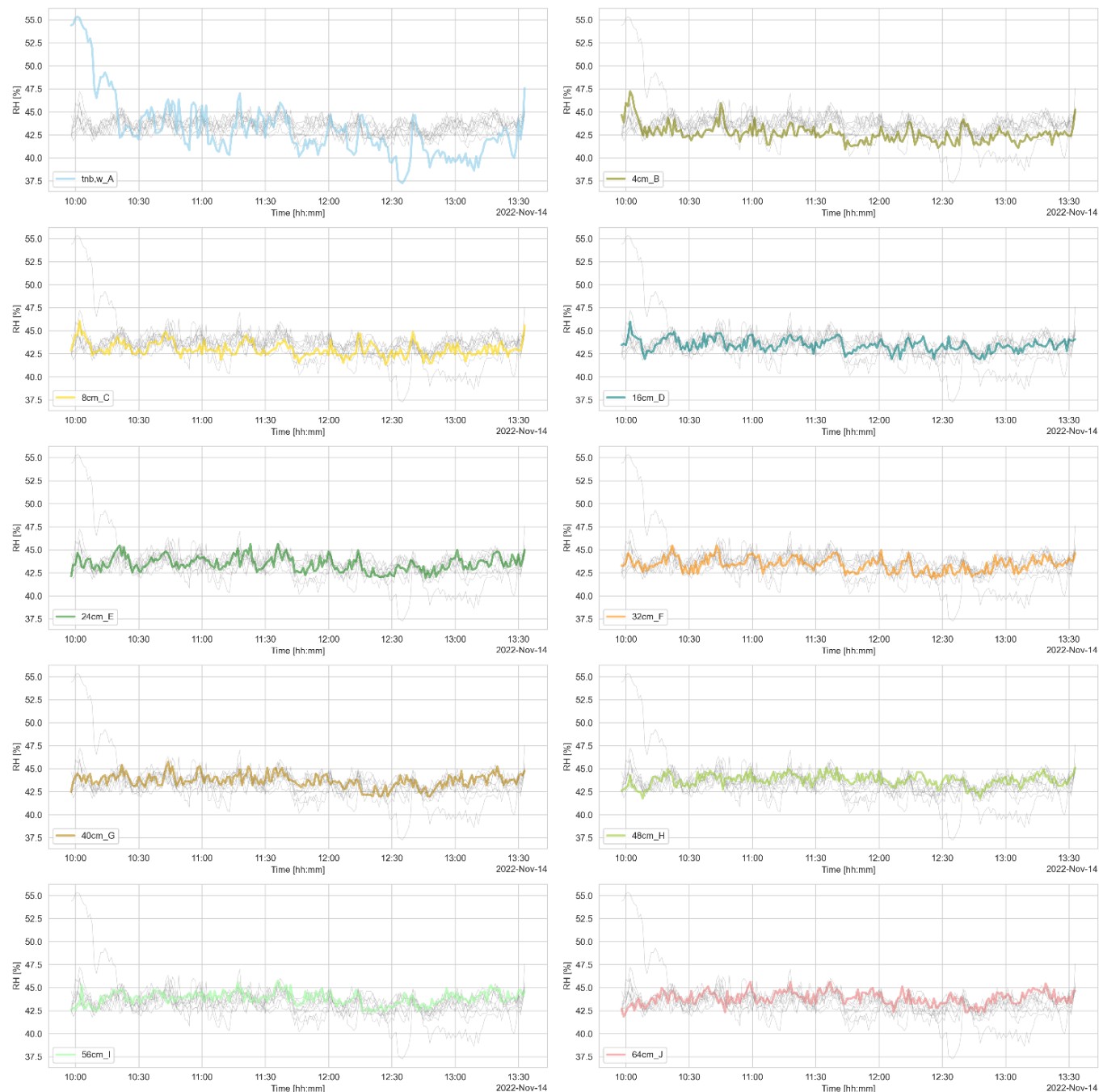

**Figure 13.** RH data—comparison in infield summer scenario.

### 3.5. Thermal Comfort Index Comparison

Considering the trend of the temperature and humidity curves and the data set, as described in the materials and method section, it is then possible to derive the PMV trends at different distances in autumn (Figure 14) and in summer (Figure 15). In both scenarios, the PMV index calculated with the data at $t_{nb,w}$_A is higher than the others calculated at the greatest distance of approximately 0.4. In terms of the PMV, the average difference is not so pronounced with data at 4 cm if compared with other curves for the greatest distances.

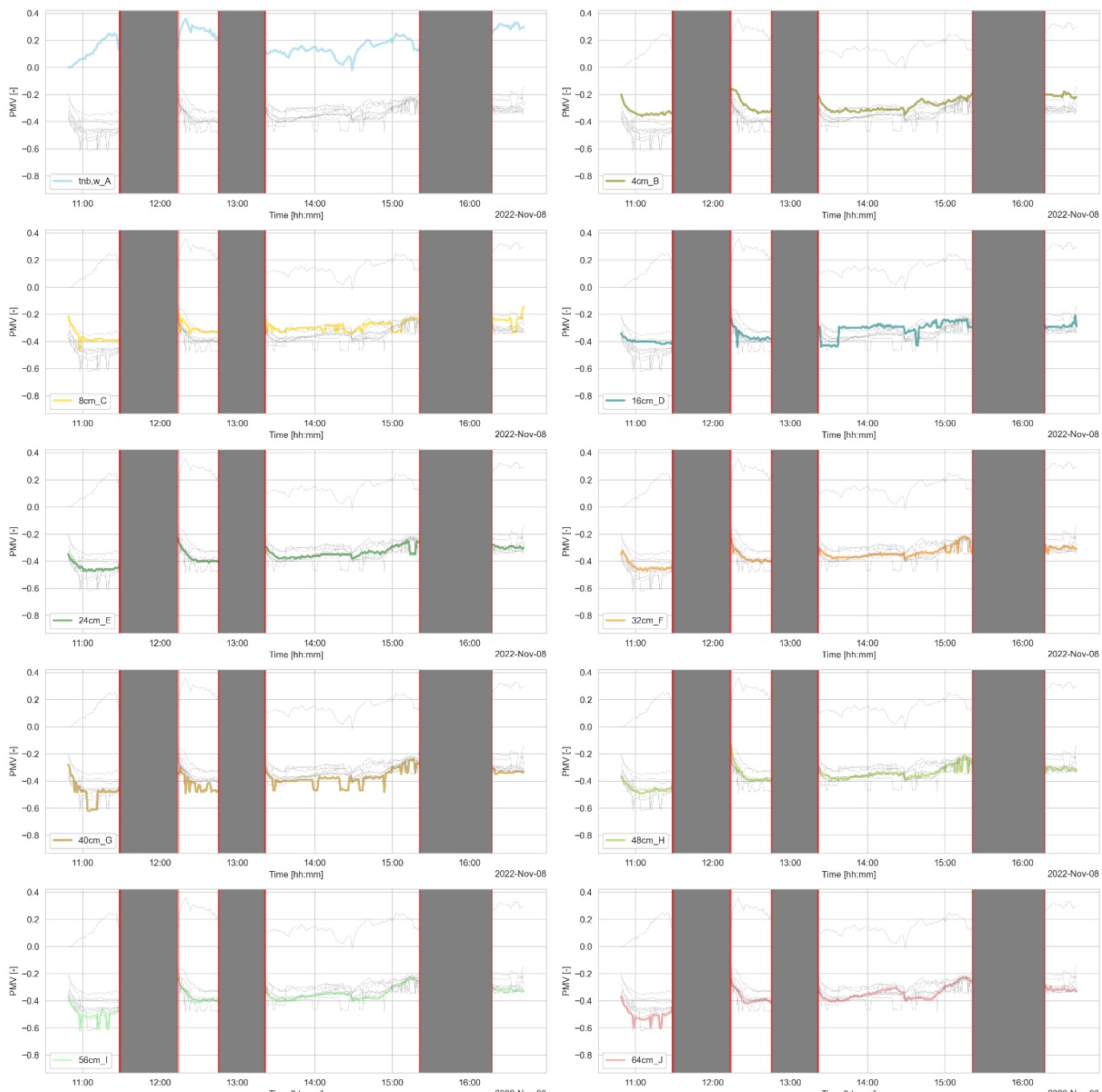

**Figure 14.** PMV data—comparison in infield autumn scenario.

The PMV trend in the simulated summer scenario (Figure 15) is strongly influenced by the pseudosinusoidal variation in air temperature, which is due to two simultaneous effects: on the one hand is the heating system, which heats the room to the desired temperature and then turns it off, and, on the other hand is the heat recovery device, which supplies fresh air to the room. The average PMV at $t_{nb,w}$_A is higher than the others calculated at the greatest distance of approximately 0.4.

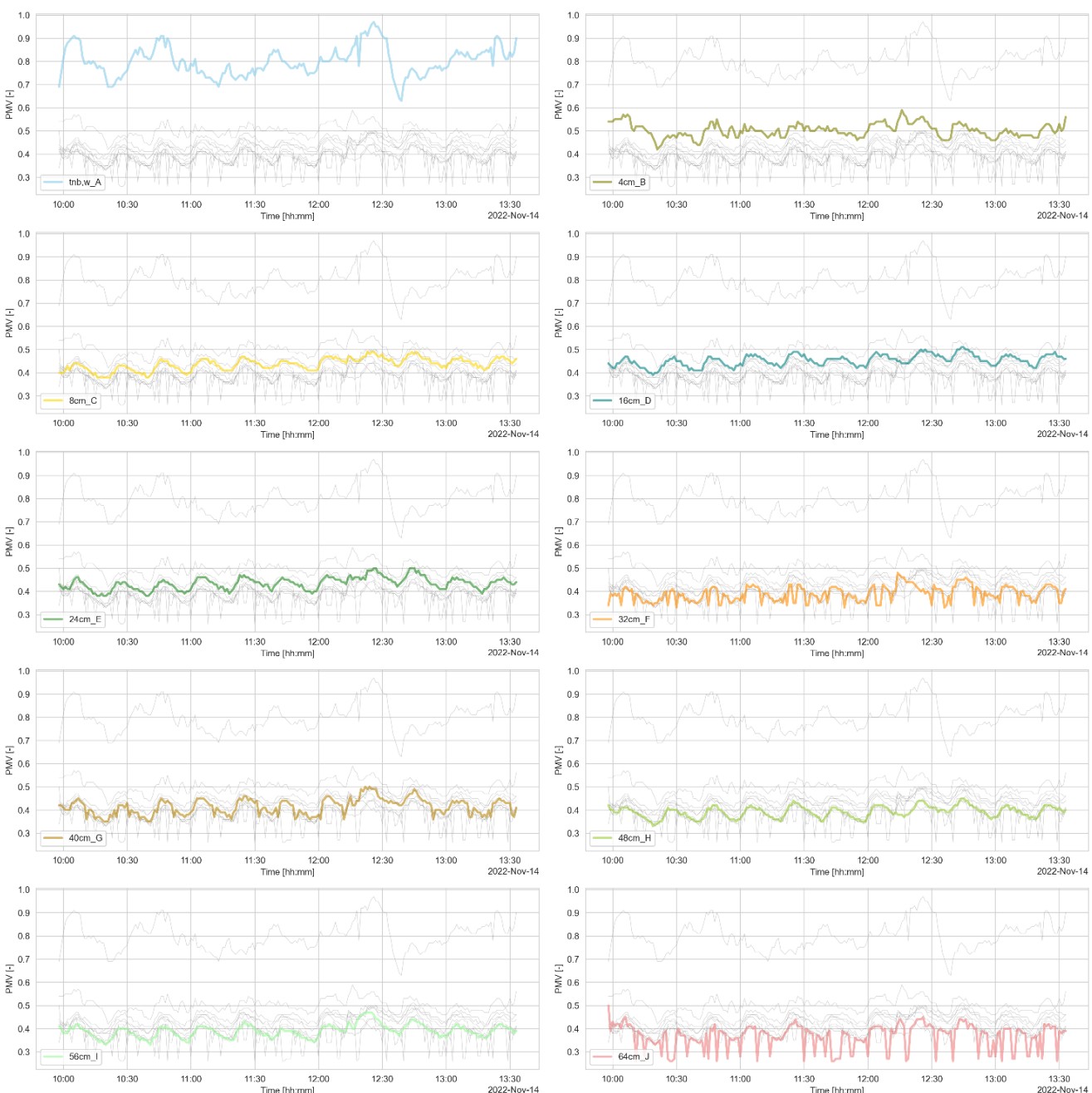

**Figure 15.** PMV data—comparison in infield summer scenario.

## 4. Discussion

This study focuses on the characterization of the SP of the thermal environmental factor using a DIY system. This test was conducted with the main objective of identifying a location where a thermohygrometer could be installed in a new wearable device. However, looking at the results of this study, it is also possible to initiate some other discussions. Namely, there are two main points of discussion: how this SP could affect the design of wearables, and how these results could also be used in conducting test with real participants in living laboratories.

### 4.1. Thermal Domain Assessed with Wearables in Field Studies

On the one hand, in the field of thermal comfort assessment with wearables, mainly field studies are conducted. The possibilities offered by the literature vary [60], and when reviewing all the studies, it is clear that, in different cases [22,61–63], technical details are

provided about the metrological characteristics of the different sensors used. In one case, a specific position [64] was used to assess heat exposure using thermometers attached to workers' shoes, and the authors emphasize how useful this position is in identifying heat exposure in the workplace. As far as we know, in all cases where the focus is on assessing thermal comfort, there are no details or discussions of where these sensors are placed and why this location has been considered. Moreover, given the previously mentioned findings, this aspect cannot be neglected. With this in mind, it is advisable to use at least two thermohygrometers: the former at a very close proximity to the human body, close to zero, so that a new concept of thermal comfort assessment can be considered and carried out, which can also take into account the HTP; the latter can be placed at a distance of at least eight centimeters in order to evaluate the thermal comfort, taking into account the air temperature and relative humidity which are not disturbed by the HTP.

The main limitations of this study are that, as the test was conducted with a single user in only two scenarios, it is not possible to extend the results. In addition, as a further limitation but at the same time as a possible cue for future fields of use, we did not consider different rods placed radially to monitor the circular gradient around the person. However, the great novelty of the proposed study lies in the definition of a procedure and easy-to-reproduce DIY system that is able to verify in different circumstances the SP due to the human body being immersed in the environment and to quantify how this presence affects the air temperature and relative humidity the further we move away from the human body.

Another limitation is that we have only considered one position—the belt where we want to install part of our wearable system. Again, this limitation is easily overcome thanks to the DIY system and the ease of replication, so that other researchers can check the SP on a different part of the human body where they want to place their wearables. To this end, in keeping with the DIY philosophy, we have decided to also share the 3D-printed parts so that anyone can replicate the study or customize it to their specific needs [65].

### 4.2. Thermal Comfort Assessed in Laboratory Studies

If, on the one hand, the use of wearables in field studies has the main limitation of not being able to control and monitor all areas exposed to environmental variables, then, on the other hand, the use of professional devices in laboratory studies has the limitation of only being able to consider a limited exposure time [28], with possible noises in terms of the so-called Hawthorne effect [66,67]. However, the latter has the advantage of being able to control all variables around the users. Indeed, the definition of quality criteria for multi-domain indoor studies [28] clearly states that measurements should be made close to the occupant, whenever possible, based on the recommendations of the domain-specific guidelines, in order to correctly assess the effect of an environmental stimulus on another domain perception or behavior, as these are the actual environmental conditions affecting the occupant. The same critical review [28] reports that the location of measurements was the only information that was reported more frequently, in 66% of the studies. Environmental measurements near the participants were more common in laboratory studies than in field experiments, where sensors were usually used to measure the average indoor conditions. In terms of thermal domain, in most studies, the height at which the thermohygrometers were installed was set according to the specifications of the ISO 7726 Standard. However, it is common to read expressions such as "in close proximity to the user", which are usually used to express the horizontal spatial distribution of this type of sensor. As it turned out in the approach followed in this study, the definition of the height at which the thermohygrometers could be installed was not sufficient. To ensure the reproducibility of the results, it could be useful to clearly indicate the distance from the human body where the thermohygrometers are installed.

For the above reason, the use of wearables has a double importance, both in the field and in the laboratory, as they always offer the possibility to monitor the SP of the ambient air temperature and relative humidity.

As a future improvement, it would also be interesting to extend the approach taken so far to users who have the ability to provide feedback on thermal perception, so that the feedback can be correlated with the PMV values calculated from environmental data monitored at different distances.

## 5. Conclusions

The research study focused on testing and evaluating the influence of the distance between the thermohygrometers and the human body on the monitoring of air temperature and relative humidity. For this purpose, 10 iButtons were used, whose performance was previously checked and calibrated in a controlled environment. We also used a DIY system to position the 10 iButtons defining a method that can be easily replicated. The results show a good linear relationship between air temperature and relative humidity measured at more than 8 cm from the human body. At 4 cm, the trend of air temperature and relative humidity is not really in line with the others. At $t_{nb,w}\_A$, the trend is irregular and does not reflect what is happening at greater distances. The same can be illustrated by looking at the trends in the PMV values calculated from the monitored air temperature and relative humidity data, assuming a constant air velocity and a mean radiant temperature corresponding to air temperature, clothing thermal resistance, and metabolic rate consistent with the scenarios considered.

We can conclude that the SP between the monitoring station and the human body is a relevant aspect that cannot be neglected in both field and laboratory tests and could also be considered and included in the following version of the technical standards.

When defining a new wearable device to assess for thermal comfort, we propose to consider at least two thermohygrometers: one to measure the "waist near body temperature"; the other at a distance of at least 8 cm from the human body in order to understand whether the "classical" characterization of thermal comfort and users' perception could also be influenced by the SP.

**Author Contributions:** Conceptualization, F.S.; methodology, F.S.; software, F.S.; validation, F.S., L.D., S.S. and M.M.; data curation, F.S.; writing—original draft preparation, F.S.; writing—review and editing, F.S., L.D., S.S. and M.M. All authors have read and agreed to the published version of the manuscript.

**Funding:** This research received no external funding.

**Institutional Review Board Statement:** Not applicable.

**Informed Consent Statement:** Not applicable.

**Data Availability Statement:** Not applicable.

**Conflicts of Interest:** The authors declare no conflict of interest.

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
