# Peer review of "Effect of Spatial Proximity and Human Thermal Plume on the Design of a DIY Human-Centered Thermohygrometric Monitoring System"

_applsci, doi:10.3390/app13084967_

Round 1
Reviewer 1 Report
The manuscript is well presented; however, there are minor details to consider.
For instance, the overlapping in Figures 2 and 3 of the T average and T of the sensor makes it difficult to appreciate the differences for the hist diagram. The differences are more visible in the boxplot diagram. I suggest changing the colors to be bright and contrasting with the average T. Same for RH.
In some parts, when referring to previous studies in the field, naming the authors could improve the readability of the text (see line 140).
A photograph of the buttons DS1923 accompanying Figure 1 could be highly appreciated.
Concerning the autumn scenario results, the periods in grey where there are no subjects in the room, difficult the overall comparison since the summer scenario was complete for all the measurement periods. It would be better if both scenarios appeared in the same period.
Figures 9 and 12 were difficult to appreciate since much data is presented.
Author Response
Dear reviewer,
thank you for your feedback.
Please consider the attached file.

Reviewer 2 Report
1. I think the introduction section is too long. Several parts can be deleted, such as the description of IoT, ML, AI, which is not the focus of this study. Section 1.1 contains several pieces of information about iButton should be moved to approaches. With such a long introduction, it is hard for me to understand what challenges this work is dealing with.
2. When presenting the results in Figure 2.3.4.5, the raw data can be moved to Support Information, since the ones with linear regression is more meaningful.
3. Results should be separated from section 2, which is the material and method only.
4. In the conclusion, the author claimed that at least 2 Thermo hygrometers, what is the "closest distance"?
5. In line 348, the author found a strong correlation between different positions. I think that is obvious due to the physical nature of thermal conduction and convection. More insights need to be provided, such as what is causing the low r coefficient.
Author Response
Dear reviewer,
Thank you for your feedback.
Please consider the attached file.

Round 2
Reviewer 2 Report
Can be published after grammar or spelling check is done.